# A single nanophotonic platform for producing circularly polarized white light from non-chiral emitters

Jose Mendoza-Carreño [1,4], Simone Bertucci[2,3,4], Mauro Garbarino[2,3], Matilde Cirignano[2], Sergio Fiorito[2], Paola Lova[3], Miquel Garriga [1], Maria Isabel Alonso [1], Francesco Di Stasio [2] ✉ & Agustín Mihi [1] ✉

Direct manipulation of light spin-angular momentum is desired in optoelectronic applications such as, displays, telecommunications, or imaging. Generating polarized light from luminophores avoids using optical components that cause brightness losses and hamper on-chip integration of light sources. Endowing chirality to achiral emitters for direct generation of polarized light benefits from existing materials and can be achieved by chiral nanophotonics. However, most chiral nanostructures operate in narrow wavelength ranges and involve nanofabrication processes incompatible with high-throughput production. Here, a single nanophotonic architecture is designed to sustain chiroptical resonances along the visible spectrum. This platform, fabricated with scalable soft-nanoimprint lithography transfers its chirality to conventional emitters (CdSe/CdS nanoplatelets, CdSe/CdS quantum dots, $CsPbBr_3$, $CsPbI_3$ perovskite nanocrystals and F8BT) placed atop, achieving a high dissymmetry emission factor ($g_{lum} > 1$). The dynamics study suggests enhanced out-coupling efficiency for one helicity by the photonic structure. Finally, a white light-emitting blend containing different emitters shows simultaneous dissymmetric emission values along the visible spectrum with this chiral nanophotonic platform.

In recent years, there has been significant attention directed towards controlling the polarization state of light using nanophotonics[1,2]. Circularly polarized light is used in a vast number of applications such as imaging[3], information encryption[4], spintronics[5], chiral photodetectors[6,7], LED for displays[8,9], chiral plasmonics[10,11] or sensing in bio-applications[12,13].

Measuring and quantifying the efficiency of circularly polarized emission (CPE) is key for comparing distinct approaches. Usually, the differential emitted intensities of both left- (LCP) and right-circularly polarized (RCP) light are normalized to the total emission, which is the so-called dissymmetry emission factor $g_{lum}$, defined as:

$$g_{lum} = 2\frac{I_{LCP} - I_{RCP}}{I_{LCP} + I_{RCP}} \qquad (1)$$

where $I_{LCP}$ and $I_{RCP}$ are the measured photoluminescence intensities for LCP and RCP, respectively. The ultimate polarization state of an emitter is governed by the selection rules in electronic transitions for the excited states in the material. Usually, electric dipole-allowed transitions are prominent for the photoluminescence processes, but

---

[1]Institute of Materials Science of Barcelona ICMAB-CSIC, Campus UAB, Bellaterra, Spain. [2]Photonic Nanomaterials, Istituto Italiano di Tecnologia, Genova, Italy. [3]Dipartimento di Chimica e Chimica Industriale, Università degli Studi di Genova, Genova, Italy. [4]These authors contributed equally: Jose Mendoza-Carreño, Simone Bertucci. ✉e-mail: Francesco.DiStasio@iit.it; amihi@icmab.es

result in low $g_{lum}$[14,15]. On the other hand, magnetic-allowed dipole transitions can show higher $g_{lum}$ values but they are lowly luminescent, thus limiting their use in practical applications[14,16,17]. Alternative magnitudes to measure the overall efficiency of CPE combine both the measured dissymmetry $g_{lum}$ and brightness of the transition involved to shed light on the photophysical parameters that govern the emission processes[14]. As another option, combining emitting materials with photonic architectures sustaining strong chiroptical properties enables these transitions to achieve a high degree of circularly polarized emission[18–22]. Nanophotonics showed great potential to endow chirality to the light emitted from non-chiral lumino-phores coupled to helical filaments[23] or metasurfaces[19] in virtue of the strong chiral near fields sustained[24–27]. More recently, bound states in the continuum (BIC) have also demonstrated a great potential for circular light production[28]. BICs are localized modes with theoretical infinite lifetimes but lying at energies above the light cone that can be excited via imperfections or geometrical factors, resulting in sharp optical features[29]. Chiral BICs demonstrated high fractions of circularly polarized light[30–32] albeit in a narrow spectral range, thus resulting in unsuitable approaches for broadband chiral applications. Designing a single chiral platform that extends its chiroptical response in a broadband range has attracted the attention of many research studies in mid (MIR) and near-infrared (NIR) spectral regions[33–37]. However, the analogous principle translated into visible range remains yet elusive. Chiroptical responses working in the visible range are generally strong, although spectrally very localized, limiting their bandwidth operation[30,32,38,39]. A recent review article summarizes the efforts pursuing the advances toward a broadband chiral response[40]. Pushing broadband chiroptical responses toward visible spectral range is of great interest for the development of new light sources with induced chirality. Broadband chiral light sources are desired for the direct implementation of circularly polarized emission in color displays or encryption in information technologies. However, endowing chirality to several emitting materials simultaneously for a broadband chiral light source remains challenging. Despite the vast literature addressing CPE, only few and recent works demonstrate chiral broadband emission for color displays exploiting chemical approaches[41,42]. To achieve broadband chiral emission using nanophotonics several architectures are needed or, alternatively, a single structure sustaining a series of optical resonances across the visible spectral range.

In this work, we show how a single chiral nanophotonic platform can be used for the efficient production of chiral photoluminescence employing arbitrary emitters along the entire visible spectrum. The chiral platform is composed of a large-area hexagonal array of triskelia motifs fabricated with scalable soft-nanoimprint lithography and coated with a high-refractive index material ($TiO_2$) for enhanced chiral light-matter interaction. Up to five different emitters covering the entire visible spectrum (CdSe/CdS core-crown nanoplatelets, Poly(9,9-dioctylfluorene-alt-benzothiadiazole) (F8BT), CdSe/CdS core-shell quantum dots, $CsPbBr_3$ and $CsPbI_3$ colloidal perovskite nanocrystals) are probed to show the broadband operation of this chiral platform. Steady-state measurements show high values of circularly polarized photoluminescence (PL) for all emitters, with $g_{lum}$ values ranging from 0.4 up to 1 across the visible spectrum under different excitation sources (laser versus light-emitting diodes excitation). Also, we perform time-resolved PL measurements to obtain insights into the emission dynamics of the emitters placed on the metasurface. Finally, a solid-blend of different colloidal emitters on top of the metasurface leads to a broadband optical response of the structure, thus enabling white chiral light. This approach paves the way for using a single platform for the efficient emission of chiral white light in practical optoelectronic devices.

## Results and discussion

### Single nanophotonic platform for broadband chiral response

The chiral metasurface is fabricated via soft nanoimprinting lithography as summarized in Fig. 1a. Briefly, a glass substrate coated with a thin layer of photoresist (SU8 2000.5 Micro Chem) is pressed against a

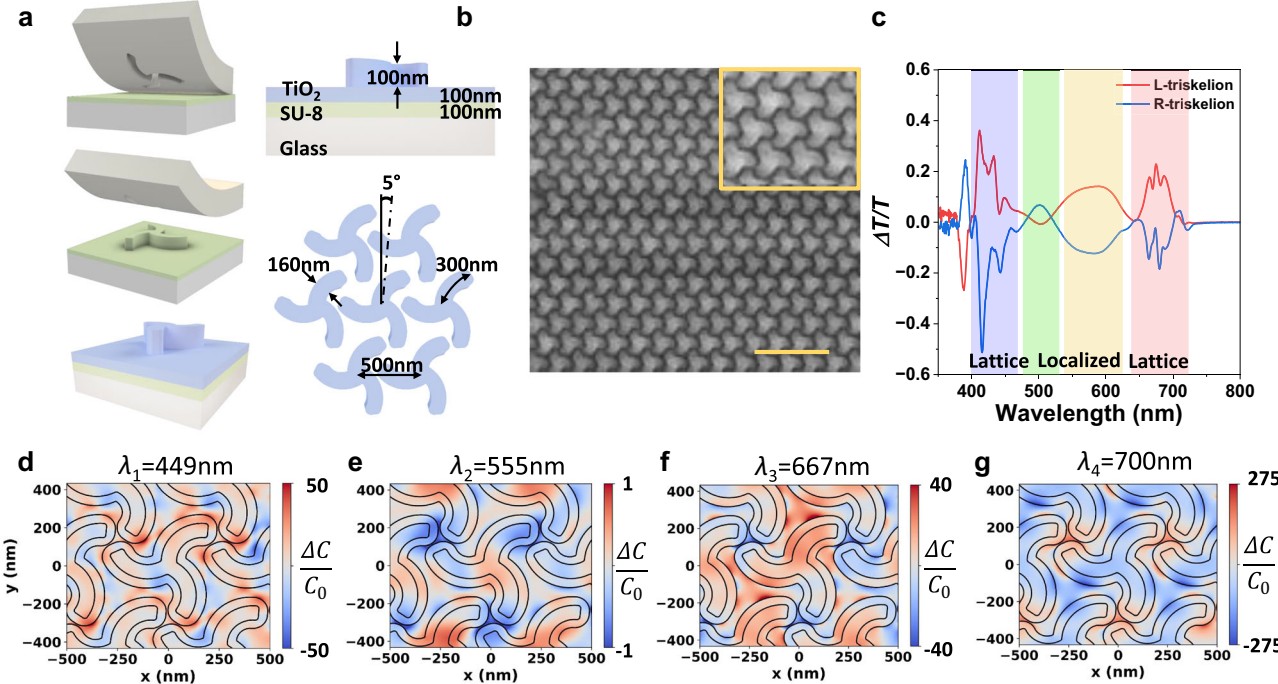

**Fig. 1 | Universal nanophotonic platform for chiral light-matter interactions.** **a** Illustration of the chiral nanophotonic platform fabrication using soft-nanoimprint lithography and the geometrical parameters. **b** Low magnification scanning electron microscopy of the $TiO_2$ triskelion chiral metasurface. Scale bar: 2 μm. The inset shows the unit cell of the chiral platform. **c** Differential circularly polarized ballistic transmittance for both enantiomorphic metasurfaces exhibiting chiral resonances along the entire visible range. Shaded colored areas indicate the different spectral regions for probing emitting nanomaterials. Net optical chirality density for an L-triskelion at the resonant wavelengths of (**d**) $\lambda_1 = 449$ nm (**e**) $\lambda_2 = 555$ nm (**f**) $\lambda_3 = 667$ nm and (**g**) $\lambda_4 = 700$ nm.

pre-patterned polydimethylsiloxane (PDMS) stamp while heating (See "Methods"). The PDMS stamp contains a left- or right-handed 2D array of triskelia. After cooling down, the PDMS stamp is removed leaving behind a corrugated SU8 film with chiral motifs followed by UV curing. The patterned photoresist is then coated with a 100 nm thick layer of titanium dioxide (TiO$_2$), a high-refractive index dielectric material transparent in the visible spectrum. High-refractive index materials have shown to be great candidates for enhanced light-matter interactions, as they can sustain both electric and magnetic resonances in the visible range[43,44]. The inter-coupling between electric and magnetic fields results in chiroptical responses at the nanoscale. These near-field effects can be further channeled to electromagnetic radiation via optical antennas, hence resulting in a boost for out-coupling efficiency[27,45].

Our chiral metasurface consists of a hexagonal array (C6 symmetry) of triskelia (300 nm arm length, 160 nm width) with a lattice parameter $\Lambda = 500$ nm and left- or right-handed orientation. The lattice parameter is optimized to sustain lattice modes in both blue (450 nm) and red (650–700 nm) regions of the electromagnetic spectrum, thus boosting the out-coupling efficiency at these wavelengths (See Supplementary Note 1 for parameters optimization). Besides, the triskelia motif is rotated with respect to the main directions of the hexagonal lattice at an angle of 5°, which according to our Finite Differences in Time Domain simulations (FDTD) results in the highest circular dichroism (CD), while minimizing the contribution from the linear polarization (Supplementary Fig. 2). Full details on the geometry can be found in Fig. 1a. A scanning electron microscopy (SEM) image in Fig. 1b shows a left-handed metasurface (triskelions with clockwise rotation) after TiO$_2$ coating (additional SEM images in Supplementary Note 2).

The optical response of the chiral metasurfaces combines large polarizabilities coming from the high-refractive index material and the triskelion design being a 2D array, thus supporting modes from the lattice and the triskelion shape. These two effects result in a rich chiroptical response along the visible range, as shown in Fig. 1c. Indeed, FDTD simulations show that, for a 500 nm periodicity, the metasurface sustains lattice modes at 450 and 700 nm, whereas the other observed resonances are associated with localized triskelion modes that can be further boosted when sweeping the lattice parameter (Supplementary Fig. 4). The experimental transmittance for both enantiomorphic metasurfaces can be found in Supplementary Fig. 1. The differential transmittance measured for left- (LCP) and right- (RCP) handed circularly polarized light presents a series of both sharp and broad resonances spanning across the visible spectrum. In order to understand the origin of each resonance, the visible spectral range is divided into 4 different regions (illustrated in Fig. 1c): the sharp transitions in the blue region at 400-450 nm, a broad and weaker green resonant band centered at 500 nm followed by the yellow-orange emission (from 550-620 nm), and finally a series of intense and sharp resonances in the red (630−720 nm). We also delve into the near and far-field response by performing FDTD simulations for the L-triskelion hexagonal array (Supplementary Figs. 2-12) by investigating the ballistic transmittance $\Delta T/T$ (Supplementary Fig. 3) and the corresponding spatial distribution of the electromagnetic fields (Supplementary Figs. 5-8). From them, we calculate the normalized optical chirality density factor $\hat{C}$[18]:

$$\hat{C} = -\frac{c\mu_0}{2}\mathrm{Im}\{\mathbf{E}^* \cdot \mathbf{H}\} \qquad (2)$$

Figure 1d-g shows the values of the normalized net optical chirality density $\triangle\hat{C}$ for four resonant wavelengths at $\lambda_1 = 449$ nm, $\lambda_2 = 555$ nm, $\lambda_3 = 667$ nm, and $\lambda_4 = 700$ nm obtained through FDTD. The $\hat{C}$ parameter accounts for the helicity of the electromagnetic field pattern compared to that of a propagating RCP wave in free space. The normalization carries information about the helicity of the

electromagnetic field distribution, assigning positive values for RCP and negative for the opposite-handedness LCP. For perfectly LCP and RCP propagating waves, the optical chirality factor is limited to −1 to +1, while a value of 0 indicates linear polarization. However, non-propagating electromagnetic fields are not restricted to these values and may exceed unity, which are known as superchiral light fields[46]. Symmetric structures can also sustain chiral near-field distributions, but their scattered field helicity must switch when changing the incoming polarization state from each handedness. Therefore, the resulting net optical chirality density when adding up both contributions is counterbalanced and the overall near-field is achiral when integrated through space. On the other hand, chiral near fields do not necessarily cancel out when switching the incoming polarization impinging upon chiral motifs, hence evolving in an uneven distribution of net optical chirality density in the near field. We compute the net optical chirality density, described in Eq. (3), for the scattered field of an L-triskelion when impinging with LCP and RCP as excitation sources for the resonant wavelengths, as shown in Fig. 1d–g.

$$\Delta\hat{C} = \hat{C}_{\mathrm{RCP}} + \hat{C}_{\mathrm{LCP}} \qquad (3)$$

In the blue spectral region at $\lambda_1 = 449$ nm (Fig. 1d), we observe strong net optical chirality hotspots up to 50-fold located in the air gaps between the triskelion motifs arrays. These regions between the triskelia motifs are the locations in which some of the emitters accumulate, especially the nanocrystals, see Fig. 2 and a close-up in Supplementary Figs. 15,16. The sign of the near fields indicates that the net optical chirality density favors in-coupling of RCP helicity (indicated as red color) for this wavelength, hence suggesting a dissymmetric preferential emission polarization of LCP due to the reciprocity principle[47].

Next, the broad resonance is assigned to the green region ($\lambda_2 = 555$ nm) at the center of the visible spectrum. Both injected polarization handedness lead to similarly scattered near-field distributions (Supplementary Fig. 6), resulting in smaller net optical chirality distribution (Fig. 1e). Therefore, we expect herein the smallest values of chirality transferred to the chiral emission. The third studied region (shaded in yellow) corresponds to a broad resonance with a maxima/minima at $\lambda_3 = 667$ nm. The net optical chirality values observed at this wavelength reach up to 40-fold (Fig. 1f), which are comparable to those seen in the blue region of the spectrum. Finally, the deep red part of the spectrum is represented by the chirality distribution at $\lambda_4 = 700$ nm (Fig. 1g). The optical chirality enhancements sustained in this spectral region surpass all the others, achieving up to 275-fold. The overall handedness at this wavelength is governed by LCP handedness, observed as a high amount of blue regions along the metasurface. Therefore, an opposite RCP preferential emission is expected at this wavelength. However, local values do not account for the entire interaction, thus integrated values are also required to assign both the overall magnitude and sign of the helicity at each wavelength. Supplementary Fig. 11 describes the in-plane integrated net optical chirality along the propagation z-axis for the resonant wavelengths studied herein, where we observe near-unity values for blue and orange wavelengths, and superchiral regime for the deep red wavelength ($\hat{C}>5$ at the active area between $200 < z < 300$ nm). As reported elsewhere, regions of higher optical chirality density are located in high-refractive index regions and interfaces[18,48]. Alternatively, we also study the integrated electric field intensity for both injection polarizations for the resonant modes both in-plane and along the propagation axis to predict the preferential emission based on reciprocity (Supplementary Figs. 9, 12)[47]. This, along with the local large optical chirality values indicates that the structure proposed herein will present its peak performance in the red part of the spectrum, as shown later in this work.

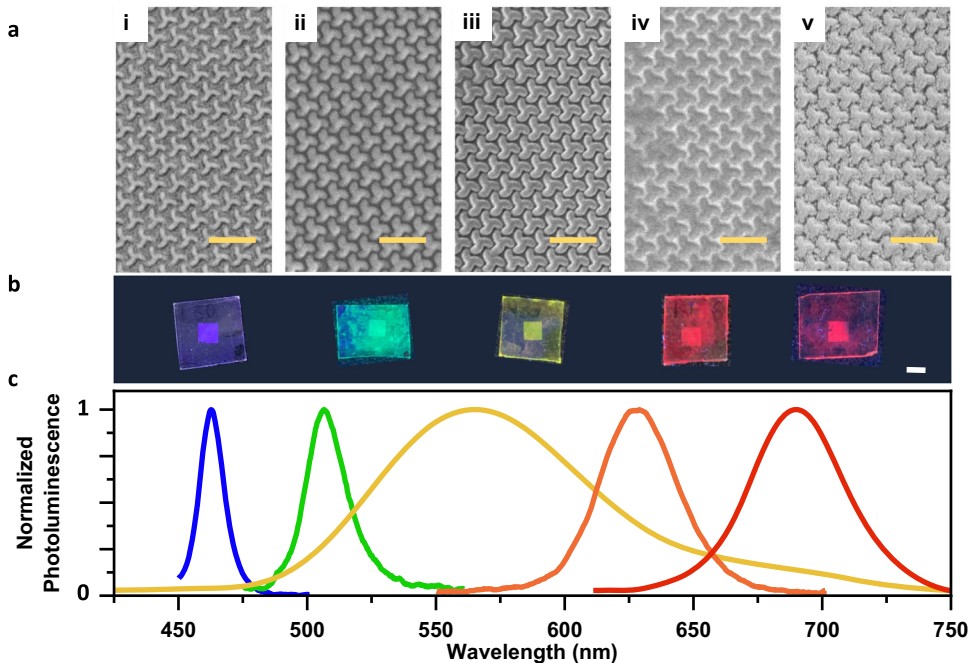

**Fig. 2 | Conventional emitters for broadband chiral light-emission.** The morphological and optical properties of the different materials used to probe the chiral nanophotonic platform. **a** scanning electron microscopy of L-triskelion chiral metasurfaces coated with, from left to right, CdSe/CdS core-crown nanoplatelets, CsPbBr$_3$ perovskite nanocrystals, conjugated polymer F8BT, CdSe/CdS core-shell quantum dots and CsPbI$_3$ perovskite nanocrystals. Scale bar: 1µm (**b**) macroscopic picture of the large-area coated metasurface under UV light exposure. Scale bar: 4 mm (**c**) normalized photoluminescence spectra covering the entire visible range.

## Circularly polarized production from conventional emitters

The abundance of chiral resonances along the visible range is investigated herein with a variety of non-chiral fluorophores of different natures, going from colloidal inorganic nanomaterials to organic emitting semiconductor molecules covering the visible spectrum (Fig. 2). The emitters are selected so their PL match the chiral resonances highlighted in the different areas shaded in Fig. 1c. From blue to red-emitting materials, we choose: CdSe/CdS core-crown nanoplatelets (NPLs), CsPbBr$_3$ perovskites nanocrystals (PNCs), F8BT conjugated polymer, CdSe/CdS core-shell quantum dots (QDs) and CsPbI$_3$ PNCs (TEM and further details on the materials available in the Supplementary Note 2). The use of fluorophores of different compositions of both organic and inorganic nature demonstrates the versatility of the nanophotonic platform here presented, indicating that chiral photoluminescence can be obtained with a large variety of different emissive materials and therefore establishing the potential of the metasurface for chirality transfer from a wide class of emitters in the visible.

Two different deposition methods are used to coat the chiral nanophotonic metasurface with the luminophores. On one hand, conventional spin-coating is used to deposit CdSe/CdS NPLs and F8BT polymeric films. On the other hand, solvent-assisted self-assembly is employed for CdSe/CdS QDs and both green and red-emitting PNCs for a better conformal coating and to avoid wasting material. Further details are found in the "Methods" section.

Figure 2a shows a high-magnification SEM of the surface of the L-triskelion metasurface covered with the corresponding active material. All the fluorophores cover entirely the metasurface, where the triskelion array is clearly distinguishable. Figure 2b shows photographs of the different metasurfaces covered with each emitter under UV light with the 16 mm$^2$ patterned area located at the center of the substrate. We work in low concentrations to maintain the optical properties of the chiral metasurface. Further details about the effect of the emitters' layer can be found in Supplementary Fig. 17. The PL of the various materials employed extends across the entire visible spectral range, as shown in Fig. 2c.

Left- and Right-handed chiral metasurfaces covered with the different luminescent materials are optically characterized in a custom-built steady-state polarization-resolved spectrometer (See "Methods"). The PL spectra and their corresponding $g_{lum}$ values for the normal direction are summarized in Fig. 3. The PL is generated under optical excitation with a 405 nm pulsed laser (200 ps pulse width at a repetition rate of 50 MHz). L-triskelion clockwise rotating arrays (Fig. 3a–e) and R-triskelion counter-clockwise arrays (Fig. 3f–j) exhibit opposite preferential polarization emission, as expected for enantiomorphic chiral architectures. This effect is clearly manifested when inspecting the dissymmetry emission value $g_{lum}$ shown in Fig. 3k–o, displaying an opposite and symmetric behavior along their emission bands.

First, blue-emitting CdSe/CdS NPLs are studied. We observe preferential LCP emission for L-triskelion metasurfaces (Fig. 3a), in agreement with the FDTD simulations. As expected, the emission switches to RCP when the enantiomorphic R-triskelion metasurface is analyzed (Fig. 3b). In this case, the dissymmetry factor $g_{lum}$ achieved reaches values of $\pm 0.3$ for both enantiomorphic metasurfaces (Fig. 3k). These large dissymmetry factor values are clearly related to the strong differential transmittance arising from the lattice resonances and the high optical chirality density observed in the near-field distributions. Few works demonstrate such large dissymmetry emission values at blue wavelengths, hence reinforcing the importance of the results achieved herein[42,49,50].

Secondly, the CsPbBr$_3$ PNCs are probed for the green spectral region. At the peak-emitting wavelength of 510 nm, the differential transmittance herein is not very high, in agreement with the moderate values of C obtained in the FDTD simulations. However, the resulting emission dissymmetry for both L- and R-triskelion (Fig. 3b, g) still results in dissymmetry values, $g_{lum}$ of $\pm 0.05$ (Fig. 3l), comparable with those obtained with chemical approaches[51].

Third, we use a conjugated fluorescent polymer, F8BT, as a probe for the yellow-shaded region. The broad PL spectrum from F8BT extends from the weak green up to the deep red regions, hence overlapping with various chiral resonances (those from 600 – 700 nm

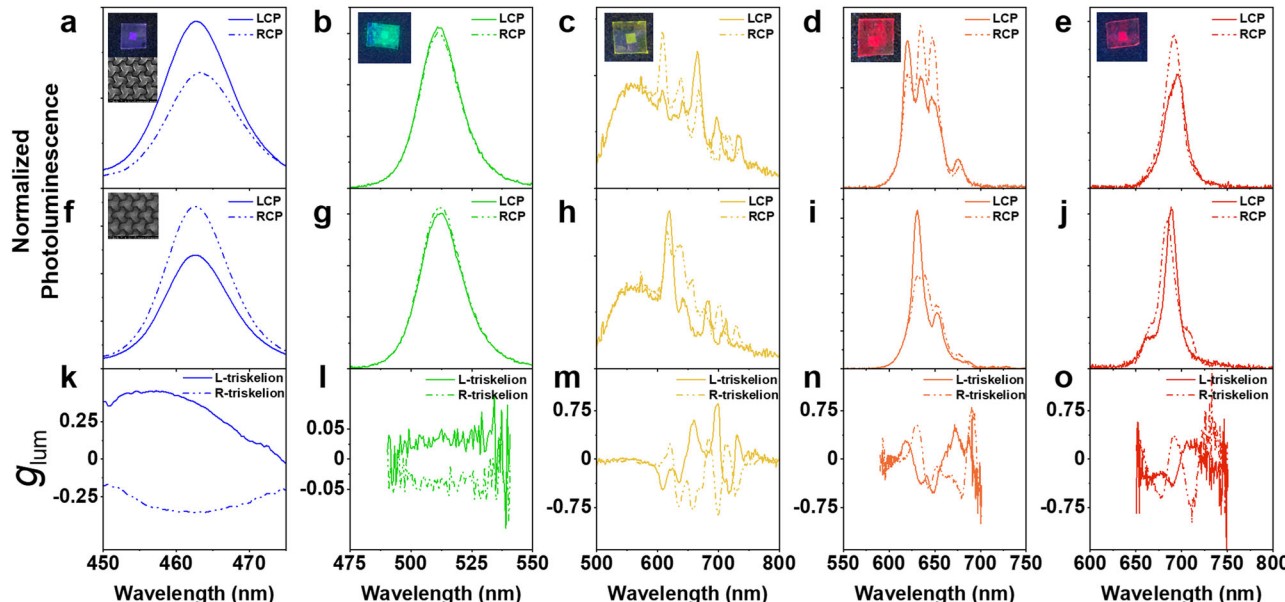

**Fig. 3 | Steady-state chiral photoluminescence in the visible range.** Circularly polarized photoluminescence for L- and R-triskelion chiral metasurfaces coated with CdSe/CdS core-crown NPLs (**a**, **f**), CsPbBr$_3$ PNCs (**b**, **g**) conjugated polymer F8BT (**c**,**h**) CdSe/CdS core-shell QDs (**d**, **i**) and CsPbI$_3$ PNCs (**e**, **j**), respectively. **k**–**o** dissymmetry emission factor $g_{lum}$ for the corresponding emitting material for both enantiomorphic chiral metasurfaces. Inset: Macroscopic image of the metasurface coated with different emitters. SEM image of L- (**a**) and R-triskelion (**f**) metasurfaces.

in Fig. 1c,d). This region of the PL spectrum shows enhanced emission peaks at various wavelengths (Fig. 3c,h), which are directly correlated to the ones observed in the differential transmittance spectrum, and they are not observed when measuring the PL outside of the patterned area (Supplementary Fig. 18). The absence of the PL enhancement for the unpatterned substrate confirms the effect of the nanophotonic platform for the efficient production of CPE. The dissymmetry factor values measured for the F8BT are much larger than the blue and green emitters. Particularly, near unity values can be attained for emission at $\lambda = 700$ nm (Fig. 3m). Therefore, the nanophotonic platform acts on the emitting material polarizing more than 50% of the PL due to an enhanced out-coupling efficiency at these longer wavelengths. The efficient angular redistribution into the normal direction of the polarized emission due to the presence of the chiral nanoantennas ends up in higher emission dissymmetry factors.

We also analyze the chiral response of the metasurface in the 600-700 nm region using two different inorganic nanocrystals with narrower emission bands. First, we study the CPE from CdSe/CdS core-shell QDs. Their PL peak emission is located at 630 nm, occupying the orange region. As seen for F8BT, new arising PL-enhanced peaks are observed when filtering the LCP and RCP components for the chiral metasurfaces at 618, 635, and 648 nm (Fig. 3d, i), achieving $g_{lum}$ values at these last two of 0.3 to −0.5 for the L-metasurface. Opposite and similar behavior is observed for the other enantiomorphic metasurface (Fig. 3n). The small discrepancies in the spectral location of the chiral emission are associated with imperfections during the nanofabrication processes that may redshift the location of the resonance. Overall the $\Delta T/T$ and the $g_{lum}$ correlate well as shown in Supplementary Fig. 19. The sign fluctuations in $g_{lum}$ perceived in some cases are attributed to the distribution and orientation of the emitters on the chiral photonic structure as illustrated in the study presented in Supplementary Fig. 21.

Lastly, we test red-emitting PNCs with peak PL at 700 nm matching the spectral features of the metasurface shaded in red in Fig. 1c. L-triskelion metasurface covered with CsPbI$_3$ exhibits preferential RCP emission at its PL peak (Fig. 3e), in agreement with the FDTD simulations. The sharp features shown in the PL for RCP (Fig. 3j) are associated with the resonant collective modes that further

outcouple into radiation, as discussed in the previous section. Nonetheless, symmetric $g_{lum}$ factors are observed for both enantiomorphs, sustaining values between 0.5-0.75 (Fig. 3o). The experimental maximum dissymmetric values obtained in the red part of the visible spectrum are in good agreement with the strong optical chirality values predicted by the FDTD simulations shown in Fig. 1g.

In addition, for both red-emitting materials, the newly arising PL peaks with preferential helicity result in large $g_{lum}$ values reaching and even exceeding ± 0.5. This effect is not observed when measuring CPE at the unpatterned area (Supplementary Fig. 18), thus reinforcing the explicit effect of the chiral nanophotonic platform. The similarities in the preferential CPE observed for all the investigated materials suggest that the polarization induction is independent of the active material and is solely related to the preferred circular polarization scattering transferred from the triskelion array used herein to the PL. For a better understanding of the mechanism taking place, in the next section we inspect the dynamical processes involved in polarization induction via time-resolved chiral photoluminescence measurements.

## Chiral light out-coupling efficiency

To disregard the pumping laser source as the cause for the dichroism in the observed PL, we shift to an incoherent optical source such as Light-emitting diodes (LEDs). LEDs are intrinsically randomly polarized, which excludes any photo-selection processes (preferential activation of emitting dipoles) induced by the optical pumping system[52].

We focus here on CdSe/CdS QDs due to their high stability conferred by the covalent bond forming the material[53,54]. However, the results obtained for CdSe/CdS QDs can be further extended to the rest of the emitting materials. Figure 4a shows the CPE of an L-triskelion coated with CdSe/CdS QDs and excited with a 405 nm LED. The metasurface exhibits a preferential RCP emission at 648 nm, similar to the case of the laser excitation (Fig. 4b). Besides, the materials used herein lack chirality in their crystalline structure, and the assembled nanomaterial is uniformly distributed in the metasurface as observed in the SEM images. The CPE signal of the metasurface indicates that the PL from the probe materials is channeled through the chiral radiation

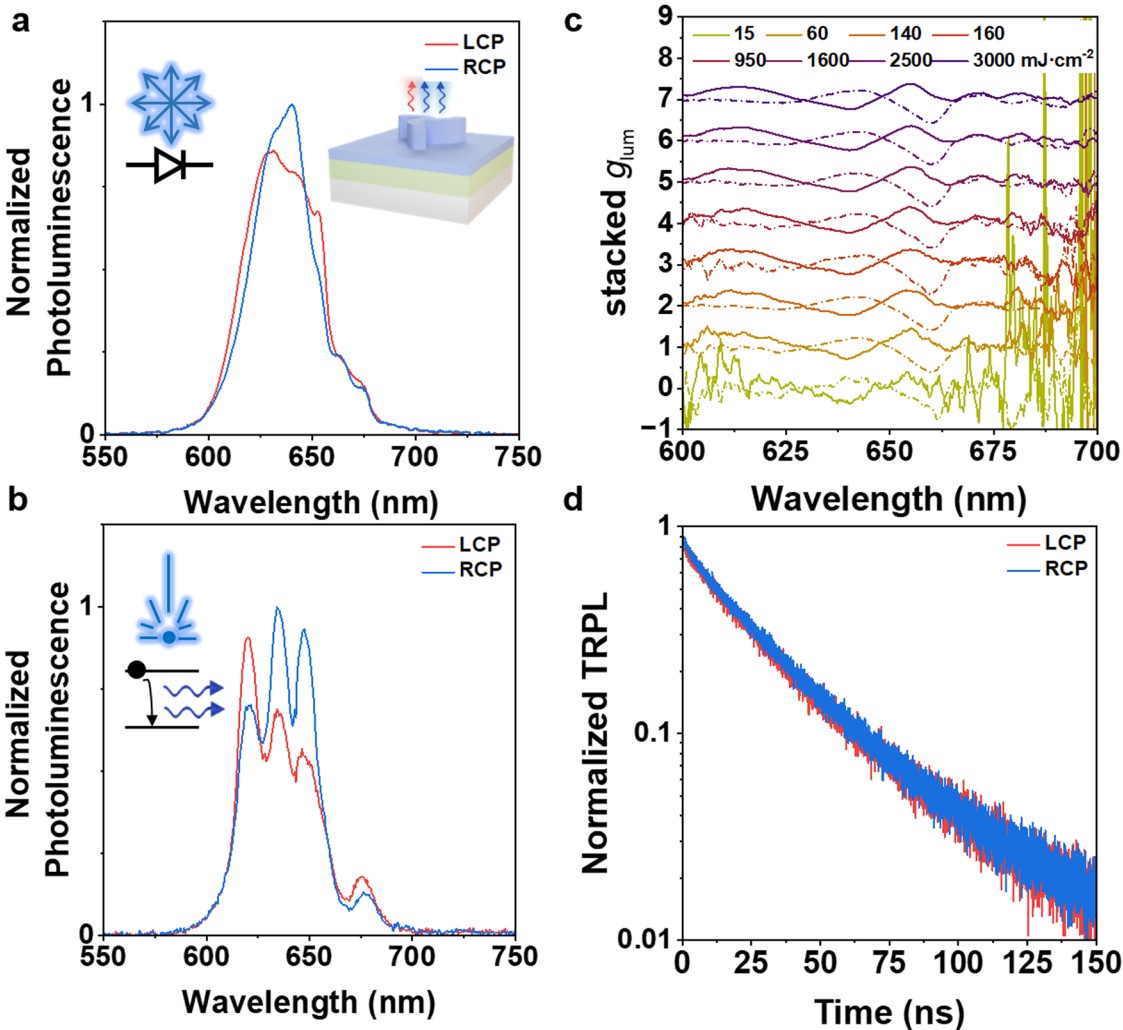

**Fig. 4 | Into the dynamics of CdSe/CdS quantum dots chiral emission.**
**a** Circularly polarized emission of L-triskelion chiral metasurface coated with CdSe/CdS QDs excited with an unpolarized LED source. Inset: 3D schematics of enhanced light-extraction efficiency for one circular polarization. **b** Circularly polarized emission of L-triskelion chiral metasurface coated with CdSe/CdS quantum dots excited with laser source. Inset: Laser source used for excitation **c** Stacked dissymmetry emission factor $g_{lum}$ for a series of power densities for L- (solid) and R-triskelion (dashed). **d** normalized time-resolved chiral photoluminescence for L-triskelion CdSe/CdS quantum dots coated metasurface.

modes supported by the triskelion antenna array, increasing the out-coupling efficiency of the system to the far-field[19,24,38,55]. Furthermore, measurements of CPE performed as a function of the LED excitation power show little variation in the dissymmetry emission factor $g_{lum}$ for the different power densities, with maximum values of up to ±0.3 and ±0.6 at 640 and 660 nm, respectively (Fig. 4c). The PL spectrum for the different power densities for L- and R-triskelion coated with CdSe/CdS QDs is found in the Supplementary Figs. 26, 27. This result opens up new possibilities for alternative strategies for chiral-electroluminescence in LED devices[8,9,56,57]. The out-coupling efficiency at normal incidence for a given polarization is higher, thus resulting in a higher probability of measuring a given handedness for the measured PL compared to its opposite helicity, as briefly schematized in inset Fig. 4a.

The differences in CPE are further investigated by measuring PL dynamics using time-resolved chiral photoluminescence (TRCPL) spectroscopy for CdSe/CdS QDs deposited on the metasurface. Previous works using plasmonic chiral metasurfaces and luminescent QDs observed coupling between the chiral absorption bands of the plasmonic metasurface and the emitting material[38]. Therefore, different up-transition rates from the ground to the excited state would result in different down-transition rates for the measured CPE. In our

case, our metasurface is composed of a lossless non-absorbing high-dielectric material (TiO₂), hence the only absorption processes at the emission wavelength are in the active material. Indeed, no emission rate differences are observed when comparing the rates in a pattern or in a flat area for any of the other emitters used herein at the chiral emission wavelengths (Supplementary Fig. 28).

We measure the TRCPL acquiring several spectra to rely on statistical differences. Specifically, for a fixed integration time of the detector, a larger number of RCP photons emitted at 648 nm are collected for an L-triskelion metasurface that sustains this preferential polarization emission (Supplementary Fig. 29). Therefore, it confirms that the scattering process excited in the chiral triskelion array by the nanocrystals shows a preferential helicity in the normal direction. However, even though we focus on a spectral wavelength range where the dissymmetry factor is large, no differences in the normalized lifetime are observed for either of the polarizations, as shown in Fig. 4d. Equivalent analysis is performed for the rest of the emitters at the resonant emitting wavelengths with no significant differences (Supplementary Figs. 30, 31). Generally, chiral metasurfaces and optical cavities can alter both radiative damping and out-coupling efficiencies of the emitters. Besides, both these modifications can be handedness-dependent. A recent study addresses a theoretical framework of

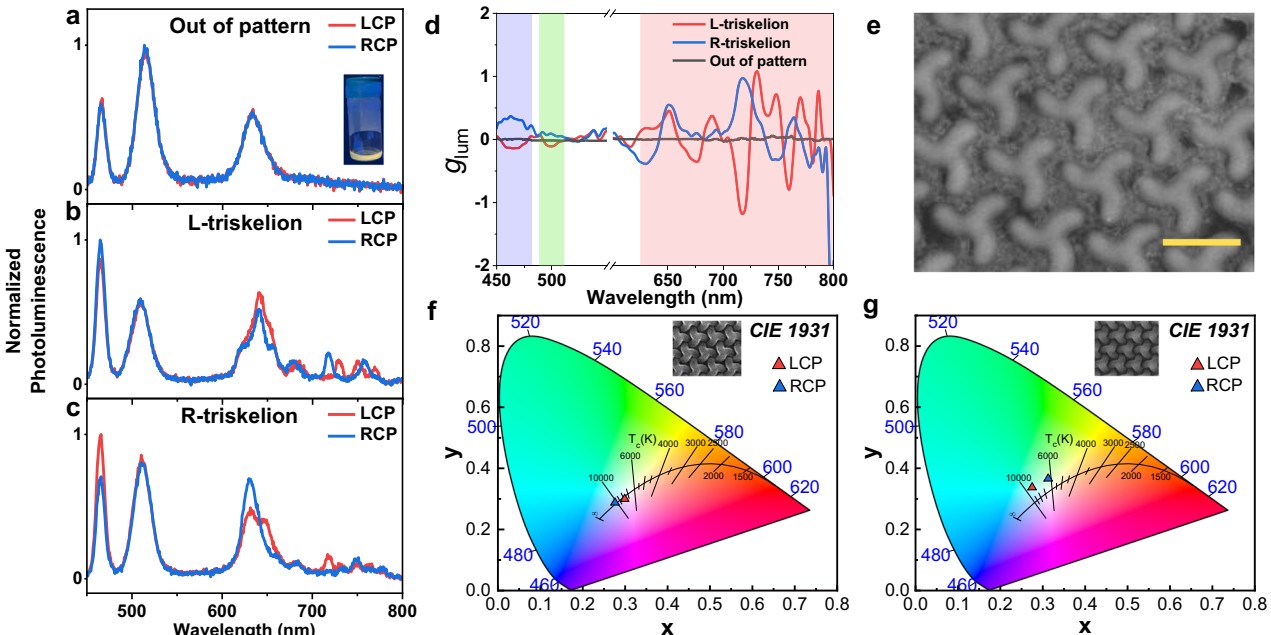

**Fig. 5 | Chiral white-light emission with a single nanophotonic platform.** Circularly polarized emission of white light for (**a**) unpatterned (**b**) L- and (**c**) R-triskelion metasurfaces. **a** Inset: white-light emissive mixed solution. **d** Dissymmetry emission factor $g_{lum}$ for the chiral white-light emission of L- (solid red), R-triskelion (solid blue) and unpatterned (solid dark). **e** Scanning electron microscopy of the deposited material on the L-triskelion chiral metasurface. Scale bar: 500 nm. CIE diagram for both circularly polarized emissions of (**f**) L-and (**g**) R-triskelion metasurfaces. Inset: Scanning electron microscopy image of L-(**f**) and R-triskelion (**g**) metasurfaces.

spin-selectivity in chiral light-matter interactions using optical cavities, where the spin dissymmetry factor is introduced to account for the increase in the local density of states[58]. However, our metasurfaces are composed of lossless dielectric materials, so we do not expect modifications in the radiative damping. In addition, we do not observe changes in the lifetime for each handedness, implying equivalent local density of optical states for both helicities. Therefore, we suggest that the polarization conversion for CPE measured in our architecture could be related to out-coupling efficiency driven by diffraction, an effect observed in non-chiral gratings[59], but in this case, operating particularly for a given polarization handedness. Besides, chiral metasurfaces have already been reported to sustain a strong angular dependence for circularly polarized emission[21,22,47]. A brief study addressing the angular emission distribution of this structure can be found in the Supplementary Figs. 22–24, where we can benefit from off-normal CPE reaching $g_{lum}$ values as high as −1.35 for CdSe/CdS QDs.

**Towards chiral white-light emission**
Finally, to illustrate the potential of our metasurfaces for chiral white-light emission, we prepare a mixed solution combining blue, green, and red emitters (CdSe/CdS NPLs, CsPbBr$_3$ PNCs, and CdSe/CdS QDs) in balanced proportions to cast a white-light emissive film (solution shown in the inset of Fig. 5a). The emitters are added gradually and the PL of the resulting solution is continuously measured until the right proportions are experimentally found. Both L- and R-triskelion metasurfaces are coated with the mixture of emitters, where each independent emitting material will experience the polarization induction corresponding to each spectral region separately. As discussed in the previous section, the effect of the excitation source is not limiting the polarization induction, hence laser excitation is used to pump the blend. As a control measurement, we measure CPE from unpatterned areas of the white-emitting blend (Fig. 5a) where no differences are observed in any of the emitting regions for the two circular polarizations, as expected. On the other hand, the emission arising from the solid blend in the patterned chiral metasurfaces

does show differences in the measured CPE for the L- and R-triskelion, respectively (Fig. 5b, c). As in the case of the individual emission, green emissive CsPbBr$_3$ PNCs sustain small amounts of differential CPE, related to the modest chiral strength of the localized resonance previously discussed. Blue emitting CdSe/CdS NPLs show comparable values of CPE as in Fig. 3a, f, showing $g_{lum}$ = 0.5 for the R-triskelion at 460 nm. In the red region of the emission spectra, where the scattering efficiency is higher, the coupled diffractive modes are visible as PL-enhanced peaks in which the preferential emission varies with the wavelength. Therefore, the dissymmetry $g_{lum}$ values in Fig. 5d switch from preferential LCP to RCP for a given enantiomorph, even reaching values of −1.2 in the case of L-triskelion at 710 nm (polarizing 60% of the emitted light). Emission localized in the deep-red region (> 700 nm) is attributed both to spectral redistribution of the emission from CdSe/CdS QDs due to the out-coupling efficiency of the metasurface and trap state emission from CdSe/CdS NPLs[60]. SEM characterization shows the uniform distribution of the colloidal emitting materials after deposition on top of the chiral metasurface (Fig. 5e). From the relative intensity of each photoluminescence colour measured in Fig. 5b and c, we can compute the International Commission on Illumination (CIE) coordinates from the filtered polarizations for L- (Fig. 5f) and R-triskelion (Fig. 5g). We show that the observed colour from the overall emission is close to white light, centered in the CIE diagram. However, due to the difference in the polarized emitted intensities for each spectral region, both LCP and RCP emissions are located in different positions in the diagram, thus obtaining different white tonalities. We observe a colder white colour tone tending to blue for the RCP emission for the L-triskelion metasurface, in contrast to warmer white LCP emission (Fig. 5f). Conversely, a warmer RCP is observed for the R-triskelion emission, as well as a colder white LCP polarization (Fig. 5g). These results prove that our single nanophotonic platform can be used in optoelectronic devices for the efficient production of white chiral photoluminescence that even presents different colour tones depending on the filtered polarization state.

In conclusion, we have designed and explored the optical properties of a single nanophotonic chiral platform for the efficient production of circularly polarized emission. The high-refractive index coated chiral metasurface, fabricated with soft-nanoimprint lithographic techniques, sustains both lattice and localized chiral resonances over the entire visible range, which enables the use of different emitting materials to probe the resonances. A strong correlation of the chiral emission dissymmetry is observed with the differential transmittance spectrum, resulting in larger differences of preferential emission in the red parts of the visible spectral range. The use of unpolarized LED excitations, as well as no differences observed in the decay rates of time-resolved chiral photoluminescence, suggest that the polarization induction is related to a larger handedness-dependent scattering out-coupling efficiency. Finally, a white light emitting mixture containing different colloidal emitters is tested for the production of a broadband chiral white-light emission with a single nanophotonic platform exceeding values of 60% CPE for the red part of the visible spectrum.

## Methods

### Sample fabrication

A photoresist (SU-8 2000.5 MicroChem 14%w) is diluted to 7%w and spin-coated onto glass substrates (Labbox Microscope slides standard line) at a speed of 3000 rpm, resulting in a thin film of about 250 nm. Next, a pre-patterned poly(dimethyl-siloxane) (PDMS) stamp is placed upon the thin photoresist layer and pressed to guarantee a conformal contact between the stamp and the substrate. The substrate is heated up to 100 °C above the glass transition of SU-8 and capillary forces drive the liquid photoresist to fill the air gaps of the pre-patterned PDMS. The substrate is left to cool down and, once the photoresist solidifies, the PDMS stamp is gently de-molded, leaving behind the patterned photoresist forming the desired chiral architecture. The patterned photoresist is exposed to ultraviolet radiation to trigger the crosslinking. To remove the excess of photo-acid generated during the crosslinking, the substrate is heated at 150 °C for 15 minutes. Finally, a 100 nm nominal thickness of titanium dioxide ($TiO_2$) is evaporated using e-beam deposition covering the architecture.

Two different strategies are used to deposit the emitting materials along the metasurface. First, traditional spin-coating techniques are used for the CdSe/CdS nanoplatelets, the F8BT, and the final mixture for the white solution. Each solution is optimized individually to obtain a uniform distribution of the emitting layer. CdSe/CdS nanoplatelets and F8BT are spin-coated dynamically at 1200 rpm and 3000 rpm respectively. The white mixture solution is spin-coated in static condition with a 20 seconds ramp to 2400 rpm. The second deposition method is a drop-assisted slow self-assembly, which consisted of coating the substrate with a drop of appropriate solvent, in our case toluene, and injecting a small amount of the emitter solution within the drop. Then, the substrates are covered with a glass crystallizer and left in a solvent-saturated atmosphere to slow down the evaporation rate and ensure the uniform assembly of the emitters atop the metasurface.

### Circular dichroism

The differential transmittance measurements are carried out in a custom-made optical setup. A white-light tungsten halogen lamp (Ocean Optics, HL-2000-HP, FL, USA) is coupled to a silver reflective collimator (RC08SMA-P01, Thorlabs) and used as an excitation light source. The light beam is passed through a Glan-Thompson Calcite Polarizer (GTH10M, Thorlabs) and directed to a super achromatic quarter wave-plate (SAQWP05M-700, Thorlabs) oriented at $\pm\pi/4$ compared to the polarization direction on a rotation mount (ELL14, Thorlabs) to obtain a circularly polarized light beam. The optical elements are automatically controlled by custom software (LabView NXG). The sample is positioned at the focal plane of a pair of 4x

objectives (RMS4X, NA = 0.1, Olympus). Finally, the transmitted light is fiber-coupled to a spectrometer (Ocean Optics, QEPro-FL).

### Steady-state circularly polarized photoluminescence

Chiral photoluminescence is characterized using the same optical elements as for the dichroic transmittance but reversed in order. First, the emitted chiral photoluminescence is directed to a quarter wave-plate (10RP52-1B, Newport) oriented at $\pm\pi/4$ and then to a linear polarizer (20LP-VIS-B, Newport), filtering one of the circular components. The excitation source consisted of a 200 ps pulsed laser source (LDH-P-C-405 laser driven with a PDL 800B driver with 5–80 MHz repetition rate)) with its wavelength peak at 405 nm.

In the case of the unpolarized excitation study, the light source used for excitation was an unpolarized LED centered at 405 nm (M405L4, Thorlabs) coupled to a 4x (RMS4X, NA = 0.1, Olympus) objective used to collimate the diverging emitted beam and refocused with an achromatic 50 mm lens upon the metasurface.

### Time-resolved circularly polarized photoluminescence

The optical set-up used for the time-resolved photoluminescence is PicoQuant Time Correlated Single Photon Counting system (Time Harp 260 PICO board, 25 ps temporal resolution; PMA Hybrid 40 detector, 250 ps response time; 405 nm LDH-P-C-405 laser driven with a PDL 800B driver with 5–80 MHz repetition rate) equipped with a compact monochromator (Solar Laser Systems). Photoluminescence lifetimes are retrieved using the PicoQuant FluoFit Pro software, and fitting the PL decay data accounting for the instrument response function.

### FDTD simulations

The FDTD simulations are performed using commercial software (ANSYS Optics). The triskelion array consists of a flat layer of photoresist (SU-8) with a refractive index of 1.6 and a hexagonal lattice of triskelion with a nominal height of 160 nm. The triskelion array is covered with a 100 nm of high-index material ($TiO2$) and its refractive index is obtained using ellipsometry measurements elsewhere[18]. The circularly polarized light is modeled as two interacting orthogonal plane waves with a phase difference of $\pm\pi/2$ injected from the glass substrate towards the air superstrate. The transmittance is measured using a power monitor placed above the triskelion array, and the near-field characterization is obtained using a 3D monitor that captures the vectorial electric and magnetic fields at the resonant wavelengths. The optical chirality density is calculated using Eq. (2) by computing the vectorial near-fields at 5 nm above the flat coating substrate XY-plane, where the emitters are positioned. To account for the net optical chirality density, we add the two different densities for the two circularly polarized excitations as specified in Eq. (3), hence resulting in areas where the circularly polarized scattered field is not compensated. This results in a chirally unbalanced environment where the emission produced therein is modified by means of the scattered far-field.

### Synthesis of emitting materials

**Materials.** Lead Bromide ($PbBr_2$, 99.99%), Lead Iodide ($PbI2$, 99.99%), Cesium Carbonate ($Cs_2CO_3$, reagent Plus, 99%), Cadmium Oxide (CdO, ≥ 99.99%), Toluene (> 99.0%), Isopropanol (> 99.8%), Ethanol (> 99.8 %), Acetonitrile (ACN, > 99%), Methyl Acetate (MeOAc 98.8%), 1-Octadecene (ODE, technical grade, 90%), Oleic Acid (90%), Oleylamine (technical grade 70%), Hydroiodic Acid (HI, 57%) and Stearic Acid (≥ 97.0%) were purchased from Sigma-Aldrich. CdO (> 99%), Se (> 99%) and S powder (99%), Trioctylphosphine (TOP, 90%) and Trioctylphosphine Oxide (TOPO, 99%) were purchased from STREM Chemicals, Propionic acid (> 99.5%) and Hexane (> 99%) from Chem Lab and Octadecyl Phosphonic Acid (ODPA, > 99%) from PCI Synthesis. All the chemicals were used without further purification.

**Synthesis of CdS/CdSe core-crown nanoplatelets.** The procedure starts with the synthesis of the 3.5 monolayers (ML) CdSe cores following a previously reported protocol[61]. In a 50 mL three-necked round-bottom flask, 679.35 mg (1 mmol) of Cadmium Stearate and 49.7 mg (0.63 mmol) of Se powder are mixed with 24 mL of ODE. Degassing is carried on for 1 h at 90 °C and then, after switching to $N_2$, the reaction batch ais heated to 160 °C and kept at that temperature for 10 min. At 215 °C, 125 uL (9.7 mmol) of propionic acid dispersed in 1 mL of ODE is swiftly injected into the flask. The reaction proceeded for 16 min at 220 °C, then heating is removed and the system is left cooling down slowly. In order to remove unreacted stearate, sample is washed at first with chloroform and acetonitrile and then with n-hexane, isopropanol and acetonitrile. Obtained cores are dispersed in 6 mL n-hexane and stored in the fridge (+ 4 °C) prior use. Following another procedure[62], in a three-neck round-bottom flask, 12 nmol of cores dispersion n-hexane, 30 μL of oleic acid and 71.8 mg (0.18 mmol) of cadmium octanoate are added to 15 mL ODE. Degassing under vacuum at 60 °C for 1 h is followed by raising T to 185 °C under $N_2$ atmosphere. 5 mL of the sulfur stock solution in oleic acid and octadecene (corresponding to 0.18 mmol of S) is added dropwise (8 mL/h) to the reaction batch and the reaction carried on for additional 10 minutes. Then, the reaction is quenched using 0.8 g of a previously prepared $Cd(OA)_2$ solution in ODE. The obtained reaction batch is diluted with toluene and obtained particles are precipitated two times with a 1:1(v/v) solution of ispropanol:acetonitrile. Finally the particles are dispersed in hexane and the dispersion is centrifuged at 4550 RCF in order to precipitate any remaining carboxylates from the reaction. The supernatant, containing the CdSe/CdS core/crown NPLs, is stored in the fridge (+ 4 °C).

**Synthesis of CsPbBr₃ nanocrystals.** The synthesis of $CsPbBr_3$ nanocubes is accomplished following a previously reported procedure[63]. First, a cesium oleate precursor solution is obtained through the mixing of 400 mg (1.23 mmol) of $Cs_2CO_3$ and 1.75 mL of oleic acid (5.54 mmol) in 15 mL of octadecene in a 40 mL vial. The vial is then heated at 120 °C under Ar flow and the reaction is carried out until all solid disappeared, then, the solution is cooled down to room temperature. Later, in a 40 mL vial, 72 mg (0.2 mmol) of $PbBr_2$, 50 μL of oleic acid and 500 μL of oleylamine are added to 5 mL of octadecene. The vial is then heated up to 175 °C until complete salt dissolution. Subsequently the reaction batch is cooled down and, at 165 °C, 500 μL of pre-heated cesium oleate precursor solution (corresponding to 0.073 mmol of Cs) are injected. Once cooled at room temperature, the solution is precipitated through centrifugation at 1574 RCF for 3 minutes. Ultimately, the particles are dispersed into 4 mL of toluene.

**Synthesis of CdSe/CdS core-shell quantum dots.** The synthesis of CdSe cores is carried out following a reported procedure[64]. First, in order to obtain CdSe cores, 280 mg of ODPA, 3 g of TOPO and 60 mg (0.467 mmol) of CdO are mixed in a 50 mL three-neck round-bottom flask and degassed at 120 °C for 60 minutes. Then, the temperature is set to 380 °C and, once obtained complete reagents dissolution, 1.5 mL of TOP are added to the reaction batch. At T = 380 °C, 600 uL of a previously prepared TOP-Se solution (corresponding to 0.72 mmol mmol of Se) are injected and reacted for 90 seconds. After cooling down to room temperature, the obtained CdSe cores are precipitated under inert atmosphere with EtOH and 3500 rpm centrifugation for 5 minutes three times and finally redispersed in toluene. Then, 1 mL of the CdSe quantum dots suspension ($1.61 \cdot 10^{-4}$ mmol) and 5 mL of ODE are added in a 25 mL three-neck round-bottom flask and degassed at 150 °C for 15 minutes and then temperature is increased to 300 °C. After, 2.76 ml of a Cd-oleate solution in ODE (corresponding to 1.38 mmol of Cd) and 2.76 mL of TOP-S solution in ODE (corresponding to 1.38 mmol of S) are mixed and slowly (5.52 mL/h) injected into the flask. At the end of the injection the reaction batch is cooled down to room temperature and washed 2 times. The first washing is accomplished by adding isopropanol and ethanol and centrifuging at 1370 RCF for 5 minutes, while the second is carried out with ethanol only. Obtained nanocrystals are then dispersed in toluene.

**Synthesis of CsPbI₃ nanocrystals**

922 mg of $PbI_2$ (0.20 mmol) are mixed in 5 mL of octadecene inside a 25 mL three neck-round bottomed flask and the system is heated at 120 °C for 1 h under nitrogen flow. Then, 0.5 mL of oleylamine, 0.5 mL of oleic acid and 0.5 mL of pre-heated stock solution of OLA-HI (corresponding to 0.35 mmol of I) are added to the flask. After 10 minutes, the temperature is increased to 260 °C. At this point, 0.5 mL of Cs-oleate (corresponding to 0.073 mmol of Cs) are injected and reacted for 60 seconds. After cooling down to room temperature, the crude solution is centrifugated at 6000 rpm for 5 minutes. The precipitate is dispersed in hexane and washed with methyl acetate at 3823 RCF for 2 minutes. Finally $CsPbI_3$ nanocubes are dispersed in hexane.

## Data availability

The experimental data can be found at: https://digital.csic.es/handle/10261/371183.

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

## Acknowledgements

This work has received funding from the Spanish Agencia Estatal de Investigación (AEI/MCIN) through grants, PID2022-141956NB-I00 MCIN/AEI/10.13039/501100011033(OUTLIGHT), and CEX2023-001263-S (Spanish Severo Ochoa Centre of Excellence program) and from the Generalitat de Catalunya (2021-SGR-00444). This research was also supported by the EIC PATHFINDER CHALLENGES project 101162112 (RADIANT), funded by the European Union. J.M.C. acknowledges an FPI fellowship (PRE2020-09411) from MICINN co-financed by the European Social Fund and the Ph.D. program in Materials Science from Universitat Autònoma de Barcelona UAB. S.B., M.C., S.F., and F.D.S. acknowledge support by the European Research Council via the ERC-StG "NANOLED" (Grant 851794). M.G. and F.D.S. acknowledge support by the European Innovation Council via the Pathfinder OPEN "TWISTEDNANO" (Grant 101046424). S.B. also acknowledges Dorwal Marchelli for his insight on spectroscopy measurements and Simone Lauciello for the HR-SEM images.

## Author contributions

J.M.C. and A.M. conceived the project. JMC fabricated the samples. JMC and SB performed the optical characterization and analyzed the data. M.G., M.C., and S.F. synthesized and provided the emitting colloidal materials. PL and SB lead the photoluminescence dynamics characterization. JMC performed the FDTD simulations. SB performed the time-resolved and white-light emission characterization. MiG and MIA designed the chiral metasurface. FDS, MIA, and AM supervised the project. All the authors contributed to the manuscript.

## Competing interests

The authors declare no competing interests.
