## [Transparent Peer Review file · Nature Communications]

A single nanophotonic platform for producing circularly polarized white light from non-chiral emitters

Corresponding Author: Dr Agustín Mihi

Version 0:

Reviewer comments:

Reviewer #1

(Remarks to the Author)

In the submitted manuscript, the authors studied a chiral nanophotonic platform that sustains chiral responses over the entire visible spectrum and thus can be potentially used for circularly-polarized (or chiral) white light emission. The demonstrated chiral structure based on L- and R-triskelion patterns utilized both lattice resonance and localized resonance to achieve significant chiral responses over the visible region. Authors developed a scalable, planar dielectric chiral structure using nanoimprint lithography and high-index material (TiO₂) deposition. Then, the planar chiral structure was coated with five different 'achiral' emitters of which emissions span the entire visible region. Moreover, authors conducted time-resolved measurements of chiral emission and identified that LCP and RCP emissions do not exhibit a difference in lifetime (Figure 4 and S20) over the entire visible range. They attribute this observation to the outcoupling efficiency enhancement from their chiral structure.

Circularly polarized luminescent materials have been actively studied. However, in many cases, they suffer from either very weak chiral response or low luminescence quantum efficiency. Therefore, chiral emission from chiral metasurfaces with conventional achiral emitters can provide a valuable alternative approach. Because most dielectric chiral structures have limitations in bandwidth, I think the current manuscript studied an important topic that can attract much attention in relevant research fields. In addition, I think the manuscript reported interesting measurements (such as time-resolved measurements of LCP and RCP emissions).

However, I think several important issues should be resolved in the current manuscript. Therefore, I'd like to ask the authors to clarify the following points.

1) The demonstrated chiral structure can exhibit significant chiral responses over the visible region. However, in the photoluminescence (PL) spectrum (both Figures 3 and 5), the handedness of chiral emission seems to change rather severely across the visible region (i.e., the sign of g_{lum} fluctuates). The differential transmittance spectrum in Figure 1c shows a more uniform chiral response by retaining the same sign of g_{lum} in most visible spectrum regions. But it seems the chiral PL spectra in both Figures 3 and 5 exhibit more fluctuations in handedness in a complicated way.

Therefore, I am wondering whether the current work indeed represents meaningful results. I expect that a more uniform sign of g_{lum} would be desirable for most cases (especially for chiral white-light emitters). I'd like to ask the authors to clarify this issue (e.g., by providing more specific application idea of the current result).

2) In both main text and supplementary information, the authors used 'optical chirality density' to describe the chiral emission behavior from their sample. They also mentioned the reciprocity principle to explain the dissymmetric chiral emission from their sample (citing Ref. [40]). However, in Ref. [40] and other related references, the generation of chiral emission is related to the 'integrated electric field intensity' in the emitter region (not optical chirality density). Because conventional achiral emitters are used in the current study (where, I suppose, light emission can be described by conventional electric dipole transitions), I think that the measured chiral PL spectra should be explained in terms of the electric field intensity (rather than optical chirality density).

I suppose this might also help understanding a discrepancy between the differential transmittance and chiral PL spectra mentioned above.

3) In the manuscript, authors argued that chiral emission is obtained from their sample because of the enhanced outcoupling efficiency in the 'normal' direction. I think this statement requires more explanations to check the validity. In fact, the patterned metasurfaces often exhibit significant angular responses. I think the authors should present the angle-resolved

spectrum (at least in simulations) to explain the angular response of their chiral structure. It would also be helpful in understanding the enhanced outcoupling efficiency in the current sample.

4) In the time-resolved measurements, authors observed that LCP and RCP emissions do not exhibit a difference in lifetime. Therefore, they attributed the observed chiral response in the PL spectrum to the enhancement of outcoupling efficiency in their chiral structure. This is an interesting observation, but it may not be true for other general cases. In general, I expect both radiative damping rate and outcoupling efficiency can have handedness dependence in chiral metasurfaces (or cavities).

For example, the following recent paper theoretically studied the spin-selectivity of the optical transition rate (or radiative damping rate) in optical cavities. Therefore, I think it would be better for the authors to add more general statements regarding this issue.

- "Spin dissymmetry in optical cavities", <https://arxiv.org/abs/2403.11358>

5) There have been studies on broadband chiral metasurfaces (based on both plasmonic and dielectric structures). I think the merits of the current approach should be clarified compared to previous studies in the literature.

For example, the following list shows a few examples:

- "Gold Helix Photonic Metamaterial as Broadband Circular Polarizer", *Science* 325, 1513 (2009).

- "Broadband plasmonic chiral meta-mirrors", *Optics Express* 31, 22415 (2023).

- "Simultaneous broadband and high circular dichroism with two-dimensional all-dielectric chiral metasurface", *Nanophotonics* 12, 4043 (2023).

- "Advances on broadband and resonant chiral metasurfaces", *npj nanophotonics* (2024), <https://doi.org/10.1038/s44310-024-00018-5>.

Reviewer #2

(Remarks to the Author)

This manuscript describes the generation of circularly polarized light by coating nanopatterned structures with different light emitters. The single platform couples to several different light emitters, and this is used to make a white light system.

The paper is well-written and interesting. A few comments:

1. The first part of the paper discusses the different effects expected in different spectral ranges based on the patterned titania surface. This isn't based on calculations that include the emitters though - how do the varying refractive index of the emitters affect this?

2. Related: It is difficult to tell from Figure 2/S13 that the particles are uniformly distributed. Similarly, Figure S10 illustrates the emitters as not coating the top of the pattern. Line 160 also states the emitters are mostly in the gaps. This should be proven experimentally.

3. It is well known that these systems also exhibit considerable variation in the angular dependence of the outcoupled luminescence. What kind of angular dependence is seen here?

4. Is there any preferential alignment of the nanoplatelets? Does this lead to dipole alignment effects? More generally, it is easy to imagine that different materials may not be randomly oriented; how does dipole alignment of the emitter affect these results?

5. Note a typo on page 14, line 255.

Reviewer #3

(Remarks to the Author)

In this research, the authors have studied a nanophotonic platform for producing circularly polarized emission in a wideband. The authors claim that conventional researches focus only on narrowband chirality. They use the nanophotonic structure to realize a white light chiral emission.

I am curious about the novelty and the presentation of this research in the following aspects.

1. Chirality in nanophotonics has been explored for decades. Many types of nanostructures have been studied before. Except the recently reported chiral BIC, almost all of them are broadband. Usually, the researchers consider only one spectral region to maximize the chiral response. In principle, most of the nanostructures without in-plane and out-of-plane mirror-reflection symmetries possess optical chirality.

2. Figures 1d-1g show that the broadband response is more like a space multiplexing. Different wavelength utilizes the chiral resonance localized at different position. For a reflection or transmission, this is pretty good. For a gain layer, this means that only the positions with maximal field can experience the chirality. This is the reason that the g factor is relatively low in this research.

3. For a broadband chirality, I would like to expect one that has a wavelength independent chiral response. In other words, the chirality should be the same or with a controllable shape at different wavelengths. Otherwise, it is no meaning to claim such a broadband. This information, of course, is not discussed and demonstrated in current research.

Version 1:

Reviewer comments:

Reviewer #1

(Remarks to the Author)

I have carefully checked the author response and revised manuscript. I think the authors addressed all of my previous comments properly by providing reasonable explanations and additional data.

They also improved the manuscript (especially, supplementary information) significantly. I have no further comments, and I think the manuscript is ready for publication.

Because most circularly polarized (or chiral) luminescent materials suffer from either very weak chiral response or low luminescence quantum efficiency, chiral emission with conventional achiral emitters via chiral metasurfaces could provide an important alternative approach. In this regard, the current manuscript has studied an important topic and reported valuable results ("broadband chiral emission in the visible spectral region"). Therefore, I believe the current manuscript could be of wide interest in relevant research fields.

Reviewer #2

(Remarks to the Author)

The authors have addressed my comments.

Reviewer #3

(Remarks to the Author)

I have carefully read the response letter and revised manuscript. The authors have properly answered all my comments. I think the current version can be considered for publication on Nature Communication.

Reviewer #1 (Remarks to the Author):

In the submitted manuscript, the authors studied a chiral nanophotonic platform that sustains chiral responses over the entire visible spectrum and thus can be potentially used for circularly-polarized (or chiral) white light emission. The demonstrated chiral structure based on L- and R-triskelion patterns utilized both lattice resonance and localized resonance to achieve significant chiral responses over the visible region. Authors developed a scalable, planar dielectric chiral structure using nanoimprint lithography and high-index material (TiO₂) deposition. Then, the planar chiral structure was coated with five different 'achiral' emitters of which emissions span the entire visible region. Moreover, authors conducted time-resolved measurements of chiral emission and identified that LCP and RCP emissions do not exhibit a difference in lifetime (Figure 4 and S20) over the entire visible range. They attribute this observation to the outcoupling efficiency enhancement from their chiral structure.

Circularly polarized luminescent materials have been actively studied. However, in many cases, they suffer from either very weak chiral response or low luminescence quantum efficiency. Therefore, chiral emission from chiral metasurfaces with conventional achiral emitters can provide a valuable alternative approach. Because most dielectric chiral structures have limitations in bandwidth, I think the current manuscript studied an important topic that can attract much attention in relevant research fields. In addition, I think the manuscript reported interesting measurements (such as time-resolved measurements of LCP and RCP emissions).

However, I think several important issues should be resolved in the current manuscript. Therefore, I'd like to ask the authors to clarify the following points.

1) The demonstrated chiral structure can exhibit significant chiral responses over the visible region. However, in the photoluminescence (PL) spectrum (both Figures 3 and 5), the handedness of chiral emission seems to change rather severely across the visible region (i.e., the sign of g_{lum} fluctuates). The differential transmittance spectrum in Figure 1c shows a more uniform chiral response by retaining the same sign of g_{lum} in most visible spectrum regions. But it seems the chiral PL spectra in both Figures 3 and 5 exhibit more fluctuations in handedness in a complicated way.

Therefore, I am wondering whether the current work indeed represents meaningful results. I expect that a more uniform sign of g_{lum} would be desirable for most cases (especially for chiral white-light emitters). I'd like to ask the authors to clarify this issue (e.g., by providing more specific application idea of the current result).

Authors: We thank the reviewer for this insightful comment. Indeed, in some of the fabricated metasurfaces, the chiral photoluminescence seems to behave in a more complex manner than the transmittance spectrum presented in **Fig. 1c**. We want to point out that **Fig. 1c** corresponds to a representative value of the $\Delta T/T$ from one of the measured metasurfaces. However, if we overlap the $\Delta T/T$ and the emission g_{lum} from the same metasurface we observe a strong correlation as illustrated in **Fig. S19b**. We take the opportunity here to show the reproducibility of the chiral optical

properties between different samples in Fig S19a, with some slight variations due to the fabrication process.

Fig. S1. a, Experimental transmittance dissymmetry factor $\Delta T/T$ of L- (solid line) and R-triskelion (dashed line) for CdSe/CdS core-crowned nanoplatelets (blue), CsPbBr₃ nanocrystals (green), F8BT conjugated polymer (yellow) CdSe/CdS core-shell quantum dots (orange) and CsPbI₃ (red). **b**, Transmittance $\Delta T/T$ (solid dark) and emission g_{lum} (solid yellow) dissymmetry factors for L-triskelion coated with F8BT conjugated polymer.

This question suggested by the reviewer encouraged us to deepen into the parameters that may influence the CPE. To provide further insights, we have first analyzed via FDTD simulations the different parameters influencing the sign of the g_{lum} summarized in a new figure Fig. S21. Namely, we have studied the influence of structural parameters that may vary within the nanofabrication process, such as the triskelion height or the coating thickness (Fig. S21a, b). We have also studied how the relative position of the emitter (simulated by dipoles at four different wavelengths) influences the resulting chiral response. We observed in these simulations how the handedness strongly depends on the location of the dipole as well as its orientation (Fig. S21c, d).

Future works could potentially address this issue and obtain *even higher values of g_{lum}* by precisely placing the emitters at specific locations within the unit cell, controlling their orientation (for instance via DNA origami) or by an improved design of the chiral geometry to offer a more homogeneous near-field distribution of chiral fields.

We have included the following modifications in the manuscript to present this study:

Page 9 of the main manuscript now reads:

“The small discrepancies in the spectral location of the chiral emission are associated with imperfections during the nanofabrication processes that may redshift the location of the resonance.

Overall the $\Delta T/T$ and the g_{lum} correlate well as shown in Fig. S19. The sign fluctuations in g_{lum} perceived in some cases are attributed to the distribution and orientation of the emitters on the chiral photonic structure as illustrated in the study presented in Fig. S21.

In the supporting information we have included the following figure:

Fig. S2 Simulation of photoluminescence emission when varying **a**, SU8 triskelion height **b**, TiO_2 coating thickness **c**, the position of the emitting dipole **d**, dipole orientation and **e**, numerical aperture of the collection system.

2) In both main text and supplementary information, the authors used ‘optical chirality density’ to describe the chiral emission behavior from their sample. They also mentioned the reciprocity principle to explain the dissymmetric chiral emission from their sample (citing Ref. [40]). However, in Ref. [40] and other related references, the generation of chiral emission is related to the ‘integrated electric field intensity’ in the emitter region (not optical chirality density). Because conventional

achiral emitters are used in the current study (where, I suppose, light emission can be described by conventional electric dipole transitions), I think that the measured chiral PL spectra should be explained in terms of the electric field intensity (rather than optical chirality density).

I suppose this might also help understanding a discrepancy between the differential transmittance and chiral PL spectra mentioned above.

Authors: We appreciate the insights from the reviewer about this topic. In this work, we use two different concepts to explain the origins of the emitted chiral light. First, we discuss the use of net optical chirality density for the resonant modes. This is the result of the different intensities for electric and magnetic field intensities induced by the electromagnetic field depending on its helicity. Second, we make use of the reciprocity principle to account for the change in handedness when referring to the emitted light. Therefore, first, we explore the excess of optical chirality induced by both wave helicities and then relate it to the opposite handedness through reciprocity. Moreover, the electric and magnetic fields for all the resonant wavelengths can be found in **Fig. S5-8** where the differences in electric field intensities depend on the emitters' location. Following the reviewer's comment, we have added two new figures in the supporting information accounting for the integrated electric field intensity for the resonant wavelengths to reinforce our statements. The first figure corresponds to the integrated electric intensity along the z-axis from the flat TiO₂ coating layer to the edge of the triskelion motif. The second figure shows the electric field intensity integrated in the XY-plane as a function of the propagation axis for the entire structure (solid lines) as well as only within the air gaps where the emitters are located (dashed lines). In both cases, the larger electric field intensity correlates with the same helicity using the optical chirality density.

Fig. S3. Integrated electric field intensity along the propagation z -axis for LCP (left) and RCP (right) waves for the resonant wavelengths **a,b** 449nm **c,d** 555nm **e,f** 667nm and **g,h** 700nm.

Fig. S4. Integrated electric field intensity for LCP (red) and RCP (blue) excitation as a function of propagation z -axis distance for **a**, 449nm **b**, 555nm **c**, 667nm and **d**, 700nm chiral resonance regions. Dashed lines correspond to the electric field intensity integrated into the air gaps applying the integration mask.

We have included a reference to these new supporting figures in the main text on page 5 which now reads:

Reference:

- [48] Phys. Rev. B 89, 045316

As reported elsewhere, regions of higher optical chirality density are located in high-refractive index regions and interfaces^{19,49}. Alternatively, we also study the integrated electric field intensity for both injection polarizations for the resonant modes both in-plane and along the propagation axis to predict the preferential emission based on reciprocity (Fig. S9, S12)⁴⁸. This, along with the local large optical chirality values indicates that the structure proposed herein will present its peak performance in the red part of the spectrum, as shown later in this work.

3) In the manuscript, authors argued that chiral emission is obtained from their sample because of the enhanced outcoupling efficiency in the 'normal' direction. I think this statement requires more explanations to check the validity. In fact, the patterned metasurfaces often exhibit significant angular responses. I think the authors should present the angle-resolved spectrum (at least in simulations) to explain the angular response of their chiral structure. It would also be helpful in understanding the enhanced outcoupling efficiency in the current sample.

Authors: As suggested by the reviewer, we include now the actual measurement of the angle-resolved circularly polarized transmittance in a new supporting information figure **Fig. S22**. We show that the angular response of the system is rather complex, with rapid sign changes around the gamma point, but we can see how at different angles the chiral response of the structure is more intense and can lead to tunable and larger values in future works focused in exploiting the angular response of this structure.

We have also added to the supporting information three new figures corresponding to the angular characteristics of the photoluminescence from a CdSe/CdS quantum dot coated triskelion array. In **Fig. S23**, we compare the emission properties of both L- and R-triskelion metasurfaces collected at 0° , 10° , 15° and 20° . In **Fig. S24**, we focus on a single enantiomeric metasurface (L-triskelion) and plot the resulting g_{lum} as a function of the emission angle, where the emission follows the diffractive orders of the grating (marked as dashed lines). Finally, we include as a control the angular characteristics of the emission from a flat film covered with the CdSe/CdS quantum dots which is absent of those features (**Fig. S25**).

To point the reader to this new study, we have modified the main text in the manuscript and included a few references already cited in our work regarding angular circularly polarized emission.

New references:

- ACS Appl. Mater. Interfaces 2023, 15, 30, 36945–36950.
- Phys. Rev. B 89, 045316
- ACS Nano 2021, 15, 8, 13781–13793

Page 11 now reads:

Therefore, we suggest that the polarization conversion for CPE measured in our architecture could be related to out-coupling efficiency driven by diffraction, an effect observed in non-chiral gratings⁶⁰, but in this case, operating particularly for a given polarization handedness. Besides, chiral metasurfaces have already been reported to sustain a strong angular dependence for circularly polarized emission^{22,23,48}. Therefore, a brief study addressing the angular emission distribution of this structure can be found in the **Supporting Information**, where we can benefit from off-normal CPE reaching g_{lum} values as high as -1.35 for CdSe/CdS quantum dots.

Fig. S5. Experimental angular transmittance for LCP (red) and RCP (blue) at (a,b) 0° and (d,e) 90° azimuthal angles. The corresponding differential transmission in each case is shown in (c) 0° and (f) 90°.

Fig. S6. Angular CPE for L- (top row) and R-triskelion (bottom row) coated with CdSe/CdS quantum dots at 0° (a, e) 10° (b, f), 15° (c, g) and 20° (d, h). The dashed dark line is a guide for the eye to follow the diffractive mode outcoupled.

Fig. S7. Dissymmetry emission factor g_{lum} for L-triskelion coated with CdSe/CdS quantum dots.

Fig. S8. Angular CPE for unpatterned emission for **a**, 0° **b**, 10° and **c**, 20°

4) In the time-resolved measurements, authors observed that LCP and RCP emissions do not exhibit a difference in lifetime. Therefore, they attributed the observed chiral response in the PL spectrum to the enhancement of outcoupling efficiency in their chiral structure. This is an interesting observation, but it may not be true for other general cases. In general, I expect both radiative damping rate and outcoupling efficiency can have handedness dependence in chiral metasurfaces (or cavities).

For example, the following recent paper theoretically studied the spin-selectivity of the optical transition rate (or radiative damping rate) in optical cavities. Therefore, I think it would be better for the authors to add more general statements regarding this issue.

- "Spin dissymmetry in optical cavities", <https://arxiv.org/abs/2403.11358>

Authors: We thank the reviewer for the interesting insights on this matter. Indeed, not all the chiral metasurfaces induce the polarization by similar manners. It was not our intention to claim such a restrictive statement regarding out-coupling efficiency to be the only mechanism that can induce handedness selectivity. We believe that, as our system is composed of lossless dielectric material, the radiative damping should not be altered. On the other hand, the local density of states for each helicity could differ as this theoretical work suggests. However, we did not observe relevant differences for the experimental lifetimes, therefore we concluded that the main phenomenon driving the dissymmetric emission was a handedness-enhanced out-coupling efficiency, as specified in our manuscript. To try and clarify this, we have rewritten the discussion regarding the out-coupling efficiency, including the suggested work.

Page 11 now reads:

“Equivalent analysis is performed for the rest of the emitting materials at the resonant emitting wavelengths with no significant differences (Fig. S19,20). Generally, chiral metasurfaces and optical cavities can alter both radiative damping and out-coupling efficiencies of the emitters. Besides, both these modifications can be handedness-dependent. A recent study addresses a theoretical framework of spin-selectivity in chiral light-matter interactions using optical cavities, where the spin dissymmetry factor is introduced to account for the increase in the local density of states⁵⁹. However, our metasurfaces are composed of lossless dielectric materials, so we do not expect modifications in the radiative damping. In addition, we do not observe changes in the lifetime for each handedness, implying equivalent local density of optical states for both helicities. Therefore, we suggest that the polarization conversion for CPE measured in our architecture could be related to out-coupling efficiency driven by diffraction, an effect observed in non-chiral gratings⁶⁰, but in this case, operating particularly for a given polarization handedness.”

New reference:

[59]: Dixon, J. et al. Spin dissymmetry in optical cavities. *arXiv.org*

5) *There have been studies on broadband chiral metasurfaces (based on both plasmonic and dielectric structures). I think the merits of the current approach should be clarified compared to previous studies in the literature.*

For example, the following list shows a few example:

- *"Gold Helix Photonic Metamaterial as Broadband Circular Polarizer", Science 325, 1513 (2009).*
- *"Broadband plasmonic chiral meta-mirrors", Optics Express 31, 22415 (2023).*
- *"Simultaneous broadband and high circular dichroism with two-dimensional all-dielectric chiral metasurface", Nanophotonics 12, 4043 (2023).*
- *"Advances on broadband and resonant chiral metasurfaces", npj nanophotonics (2024), <https://doi.org/10.1038/s44310-024-00018-5>.*

Authors: We appreciate the comments from the reviewer. To stress the relevance of this work compared with the vast literature regarding chiral nanophotonics, we have modified the introduction of this article, adding some relevant articles regarding broadband chiral metasurfaces.

In page 2, now reads:

“Chiral BICs demonstrated high fractions of circularly polarized light^{31–33} albeit in a narrow spectral range, thus resulting in unsuitable approaches for broadband chiral applications. Designing a single chiral platform that extends its chiroptical response in a broadband range has attracted the attention of many research studies in mid (MIR) and near-infrared (NIR) spectral regions^{34–38}. However, the analogous principle translated into visible range remains yet elusive. Chiroptical responses working in the visible range are generally strong, although spectrally very localized, limiting their bandwidth operation^{31,33,39,40}. A recent review article summarizes the efforts pursuing the advances toward a broadband chiral response⁴¹. Pushing broadband chiroptical responses toward visible spectral range is of great interest for the development of new light sources with induced chirality. Broadband chiral light...”

New references:

[34]: Gansel, J. K. *et al.* Gold Helix Photonic Metamaterial as Broadband Circular Polarizer. *Science* **325**, 1513–1515 (2009).

[35]: Whiting, E. B., Kang, L., Jenkins, R. P., Campbell, S. D. & Werner, D. H. Broadband plasmonic chiral meta-mirrors. *Opt. Express, OE* **31**, 22415–22423 (2023).

[36]: Wang, R., Wang, C., Sun, T., Hu, X. & Wang, C. Simultaneous broadband and high circular dichroism with two-dimensional all-dielectric chiral metasurface. *Nanophotonics* **12**, 4043–4053 (2023).

[37]: Hu, J. *et al.* All-dielectric metasurface circular dichroism waveplate. *Sci Rep* **7**, 41893 (2017).

[41]: Deng, Q.-M. *et al.* Advances on broadband and resonant chiral metasurfaces. *npj Nanophoton.* **1**, 1–22 (2024).

Reviewer #2 (Remarks to the Author):

This manuscript describes the generation of circularly polarized light by coating nanopatterned structures with different light emitters. The single platform couples to several different light emitters, and this is used to make a white light system.

The paper is well-written and interesting. A few comments:

1. *The first part of the paper discusses the different effects expected in different spectral ranges based on the patterned titania surface. This isn't based on calculations that include the emitters though - how do the varying refractive index of the emitters affect this?*

Authors: We thank the reviewer for this comment. Truly, our simulations did not consider the layer of emitters after their deposition. To account for their interaction and the final response after their deposition, we performed FDTD simulations for two different situations, based on the experimental SEM images (**Fig. S15,16**). These results for the two different emitters' dispositions is summarized in a new supporting information figure **Fig. S17**. First, we modeled the deposited nanocrystals located only at the walls of the triskelia array (**Fig. S17a**). Secondly, we consider the emitters as a uniform thin layer deposited atop the TiO₂ flat layer (**Fig. S17b**). We observe that the optical properties are maintained when working in low concentrations for both models, that is, when the thickness of the emitting layer is small compared to the triskelion height. We have noted this point in the manuscript.

Page 7 now reads:

Error! Reference source not found.b shows photographs of the different metasurfaces covered with each emitter under UV light with the 16mm² patterned area located at the center of the substrate. **We work in low concentrations to maintain the optical properties of the chiral metasurface. Further details about the effect of the emitters' layer can be found in the Supplementary.** The PL from the different materials used covers the full visible spectral range, as shown in **Error! Reference source not found.c**.

Fig. S9. Transmittance dissymmetry factor $\Delta T/T$ for emitters modeled as stick to the walls (top row) with thicknesses of a, 30nm and b, 50nm. Transmittance dissymmetry factor $\Delta T/T$ for emitters modeled as a thin layer (bottom row) for thicknesses of c, 30nm and d, 150nm.

2. Related: It is difficult to tell from Figure 2/S13 that the particles are uniformly distributed. Similarly, Figure S10 illustrates the emitters as not coating the top of the pattern. Line 160 also states the emitters are mostly in the gaps. This should be proven experimentally.

Authors: We appreciate the comments of the reviewer about the deposition properties. It is difficult to control the deposition of nanoemitters into corrugated metasurfaces, as the affinity for deposition in corrugated surfaces differs from that of flat unpatterned ones. The different type of emitters used herein formed thin films more or less homogeneous, with some of the nanocrystals accumulated in the gaps between triskelions. We have added to the supporting information a new figure **Fig. S16** with additional SEM images of high magnifications to experimentally show how the fluorophores are positioned on the chiral nanostructure. We have modified the main text to indicate these observations.

Page 5 now reads:

“In the blue spectral region at $\lambda_1=449$ nm (**Error! Reference source not found.d**), we observe strong net optical chirality hotspots up to 50-fold located in the air gaps between the triskelion motifs arrays. These regions **between the triskelia motifs are the locations in which some of the emitters accumulate, specially the nanocrystals, see Fig 2 and a close-up in Fig S15,16.**”

Regarding the uniformity of the deposition, our statement referred to the disposition of emitters along the entire metasurface rather than the density of emitters per unit cell. To clarify this statement, we have rephrased the uniformity of the deposition in terms of covering area.

Page 7 now reads:

“All the fluorophores covered **entirely** the metasurface, where the triskelion array is clearly distinguishable.”

Fig. S16: High magnification of L-triskelion metasurface coated with **a** CdSe/CdS core-crown nanoplatelets **b**, F8BT conjugated polymer **c**, CdSe/CdS core-shell quantum dots and **d** CsPbI₃ perovskite nanocrystals.

3. It is well known that these systems also exhibit considerable variation in the angular dependence of the outcoupled luminescence. What kind of angular dependence is seen here?

Authors: As suggested by the reviewer, we include now the actual measurement of the angle-resolved circularly polarized transmittance in a new supporting information figure **Fig. S22**. We show that the angular response of the system is rather complex, with rapid sign changes around the gamma point, but we can see how at different angles the chiral response of the structure is more intense and can lead to tunable and larger values in future works focused in exploiting the angular response of this structure.

We have also added to the supporting information three new figures corresponding to the angular characteristics of the photoluminescence from a CdSe/CdS quantum dot coated triskelion array. In **Fig. S23**, we compare the emission properties of both L- and R-triskelion metasurfaces collected at 0°, 10°, 15° and 20°. In **Fig. S24**, we focus on a single enantiomeric metasurface (L-triskelion) and plot the resulting g_{lum} as a function of the emission angle, where the emission follows the diffractive

orders of the grating (marked as dashed lines). Finally, we include as a control the angular characteristics of the emission from a flat film covered with the CdSe/CdS quantum dots which is absent of those features (Fig. S25).

To point the reader to this new study, we have modified the main text in the manuscript and included a few references already cited in our work regarding angular circularly polarized emission.

New references:

- ACS Appl. Mater. Interfaces 2023, 15, 30, 36945–36950.
- Phys. Rev. B 89, 045316
- ACS Nano 2021, 15, 8, 13781–13793

Page 11 now reads:

Therefore, we suggest that the polarization conversion for CPE measured in our architecture could be related to out-coupling efficiency driven by diffraction, an effect observed in non-chiral gratings⁶⁰, but in this case, operating particularly for a given polarization handedness. Besides, chiral metasurfaces have already been reported to sustain a strong angular dependence for circularly polarized emission^{22,23,48}. Therefore, a brief study addressing the angular emission distribution of this structure can be found in the Supporting Information, where we can benefit from off-normal CPE reaching g_{lum} values as high as -1.35 for CdSe/CdS quantum dots.

Fig. S10. Experimental angular transmittance for LCP (red) and RCP (blue) at (a,b) 0° and (d,e) 90° azimuthal angles. The corresponding differential transmission in each case is shown in (c) 0° and (f) 90°.

Fig. S11. Angular CPE for L- (top row) and R-triskelion (bottom row) coated with CdSe/CdS quantum dots at 0° (a, e) 10° (b, f), 15° (c, g) and 20° (d, h). The dashed dark line is a guide for the eye to follow the diffractive mode outcoupled.

Fig. S12. Dissymmetry emission factor g_{lum} for L-triskelion coated with CdSe/CdS quantum dots.

Fig. S13. Angular CPE for unpatterned emission for **a**, 0° **b**, 10° and **c**, 20°

4. Is there any preferential alignment of the nanoplatelets? Does this lead to dipole alignment effects? More generally, it is easy to imagine that different materials may not be randomly oriented; how does dipole alignment of the emitter affect these results?

Authors: We do not observe any preferential assembly when inspecting the high-magnification SEM images, as shown in both **Fig. S15** and **Fig. S16**. As we are depositing these nanoplatelets using the spin-coating technique, which is a fast process that leaves no time for self-assembly processes. We believe that our emitters are randomly oriented along the entire metasurface, therefore allowing all the dipolar direction contributions to be coupled to the far-field.

We have investigated via FDTD simulations how the different location of an emitter (simulated by dipoles of four different wavelengths) in the photonic unit cell and its relative orientation affect the chiral properties observed. The results summarized in the new **Fig. S21**, show how the handedness strongly depends on the location of the dipole as well as its orientation.

We have modified the manuscript to redirect the reader to this new study.

Page 9 now reads:

“The small discrepancies in the spectral location of the chiral emission are associated with imperfections during the nanofabrication processes that may redshift the location of the resonance. Overall the $\Delta T/T$ and the g_{lum} correlate well as shown in **Fig. S19**. The sign fluctuations in g_{lum} perceived in some cases are attributed to the distribution and orientation of the emitters on the chiral photonic structure as illustrated in the study presented in **Fig. S21**.”

Fig. S141. Simulation of photoluminescence emission when varying **a**, SU8 triskelion height **b**, TiO₂ coating thickness **c**, position of the emitting dipole **d**, dipole orientation and **e**, numerical aperture of the collection system.

5. Note a typo on page 14, line 255.

We thank the reviewer for noticing this typo. We have corrected the Supporting Information regarding the figure referencing.

Reviewer #3 (Remarks to the Author):

In this research, the authors have studied a nanophotonic platform for producing circularly polarized emission in a wideband. The authors claim that conventional researches focus only on narrowband chirality. They use the nanophotonic structure to realize a white light chiral emission.

I am curious about the novelty and the presentation of this research in the following aspects.

1. Chirality in nanophotonics has been explored for decades. Many types of nanostructures have been studied before. Except the recently reported chiral BIC, almost all of them are broadband. Usually, the researchers consider only one spectral regions to maximize the chiral response. In principle, most of the nanostructures without in-plane and out-of-plane mirror-reflection symmetries possess optical chirality.

Authors: We agree with the reviewer that chiral nanophotonics have been studied before, however most of the architectures studied operate in long wavelength ranges, such as NIR, MWIR or far beyond to MIR. Our work is novel because we show a scalable architecture, that operates in the visible region, we demonstrate its validity for a variety of light emitters and we report relevant values of g_{lum} .

To help contextualize our findings within the literature, we list below a few relevant works detailing structure, applications and wavelength bandwidth. We observe that, for NIR or larger wavelengths, the figures of merit are remarkable in a broadband range. However, when we analyze the literature regarding visible applications (shaded in blue), the results are localized in smaller spectral bands, hence limiting their implementation within broadband light sources. In this regard, our work tries to enable the use of broadband chiral applications in the visible range.

Reference	Structure	Application	Wavelength bandwidth	Dissymmetry factor	Comments
Science 325, 1513 (2009)	3D gold helix	CD/Polarizer	3-6.5 μm	$\Delta T = 70\%$ No g_{lum}	MIR
Nanophotonics, vol. 12, no. 21, 2023, pp. 4043-4053	Trapezoidal (C2)	CD/Polarizer	1.35-1.7 μm	$\Delta T = 80\%$ No g_{lum}	MWIR
Sci Rep 7, 41893 (2017)	Z-shaped	CD/Waveplate	1.35-1.7 μm	$\Delta T = 97\%$ (1.5 μm) No g_{lum}	MWIR
Phys. Rev. Lett. 106, 057402	Gammadion (C4)	CPE	1000-1100nm	$g_{lum} = 0.52$	MWIR. Changes sign of CPE
Phys. Rev. B 89, 045316	Gammadion (C4)	CPE	900-950nm	$g_{lum} = 1.6$	NIR

Sci Rep 5, 13034 (2015)	U-shaped	CD	850-1150nm	$\Delta T = 50\%$ No g_{lum}	NIR. Extrinsic chirality
Adv. Optical Mater. 2023, 11, 2300197	Tilted bars	CPE	975-1050nm	$g_{lum} = 1.48$	NIR
ACS Nano 2021, 15, 8, 13781–13793	Nanotriangles (C3)	CPE	750-850nm	$g_{lum} = 1$	NIR. Off-normal extrinsic chirality
Laser & Photonics Reviews 2019, 13, 1800276	Nanoslits (C4)	CD/CPE	600-650nm	$g_{lum} = 0.52$	Visible
Science 377,1215-1218(2022)	Tilted and slanted bars	CPE	612nm	$g_{lum} \sim 2$	qBIC
Light Sci Appl 7, 17158 (2018)	Gammadion (C4)	CD	510-560nm	$\Delta T = 80\%$ No g_{lum}	-
Laser Photonics Rev. 2023, 17, 2200611	Tilted bars	CPE	550-560nm	$g_{lum} = 1.8$	qBIC
Adv. Mater. 2023, 35, 2210477	Gammadion (C4)	CPE	500-550nm / 600-680nm	$g_{lum} = 0.3$	-
This work	Triskelion (C3)	CPE	400-700nm	$g_{lum} = 1.2$	Broadband in the visible range

2. Figures 1d-1g show that the broadband response is more like a space multiplexing. Different wavelength utilizes the chiral resonance localized at different position. For a reflection or transmission, this is pretty good. For a gain layer, this means that only the positions with a maximal field can experience the chirality. This is the reason that the g factor is relatively low in this research.

Authors: We respectfully disagree with this comment. Our measurements indicate that we obtain both good chiroptical response in transmission but also, we are able to exploit those intense chiral fields to produce remarkable values of CPL. Even in the case of “space multiplexing” suggested by the reviewer, the g_{lum} values obtained are not low compared to similar systems reported in the literature (see for example Laser & Photonics Reviews 2019, 13, 1800276, reported a g_{lum} of 0.52). In fact, we obtain large g_{lum} values of 0.5 for CdSe/CdS core-crown nanoplatelets at 460nm, a considerable enhancement compared to the only reported value of 5.29×10^{-4} in literature for this material (Adv. Optical Mater. 2021, 9, 2101142). Besides, we obtain $g_{lum} = 0.85$ at $\lambda = 700\text{nm}$ for F8BT, above the reported values in the literature.

- $g_{lum} = 0.57$ in ACS Appl. Mater. Interfaces 2020, 12, 35, 39471–39478

- $g_{lum} = 0.64$ in Light Sci Appl 8, 120

- $g_{lum} = 0.5$ in Adv Mater. 2013 May 14; 25(18): 2624–2628

For CdSe/CdS quantum dots, the values obtained in this work range from 0.5 up to 1.35 at $\lambda = 630\text{nm}$ by benefiting from off-normal emission, which is 3 orders of magnitude larger than the values found

in previous works for this material (4.66×10^{-4} in *ACS Nano* 2018, 12, 6, 5341–5350). Finally, for perovskite nanocrystals, we obtain values up to $g_{\text{lum}} = 0.75$ for CsPbI₃ at $\lambda = 700\text{nm}$ at room temperature, which is much higher than the values obtained in literature for this material:

- $g_{\text{lum}} = 0.34$ in *Adv. Mater. Interfaces* 2023, 10, 2300576
- $g_{\text{lum}} = 0.2$ at $T = 80\text{K}$ in *Nano Lett.* 2022, 22, 10, 3961–3968
- $g_{\text{lum}} = 0.3$ in our previous work in *Adv. Mater.* 2023, 35, 2210477

In sum, we show in our work a single platform providing high values of g_{lum} for achiral materials such as CdSe/CdS core-crown nanoplatelets and quantum dots, and we obtain similar and higher g_{lum} values for already reported materials such as F8BT and perovskite nanocrystals.

3. For a broadband chirality, I would like to expect one that has a wavelength independent chiral response. In other words, the chirality should be the same or with a controllable shape at different wavelengths. Otherwise, it is no meaning to claim such a broadband. This information, of course, is not discussed and demonstrated in current research.

Authors: Broadband chirality is not straightforward to obtain. Most of the broadband chiral responses are located in infrared and longer wavelengths due to the required sizes of the scattering building blocks (detailed in *npj Nanophoton.* 1, 20 (2024)). The examples provided in this recent review focused on broadband and chiral resonances with the absence of peaks operate above the visible range:

- 1.2-2 μm in *Opt. Express* 26, 31484-31489 (2018)
- 1.2-2 μm in *Nanotechnology* 31 295203
- 5-5.5 μm in *Opt. Lett.* 45, 5372-5375 (2020)
- 1.3-1.8 μm in *Nanophotonics*, vol. 12, no. 21, 2023, pp. 4043-4053.
- 7-25 μm in *RF Microw Comput Aided Eng.* 2021; 31:e22693

The reported references in the visible range either have a smaller operational bandwidth or are not planar metasurfaces, but are rather complicated 3D-stacking.

- 500-600 nm in *Physical Review Letters* 125, 267402 (2020)
- 600-1200 nm in *Nat Commun* 3, 870 (2012)

In our work, we report a chiral response from 400nm up to 750nm using a single planar chiral architecture. We understand that the term “broadband” is very general and might have different expectations for different readers. However, there are many nice works in nanophotonics using the term broadband. For instance, “broadband” was used for a 200 nm range here: “Broadband highly directive 3D nanophotonic lenses, *Nature Communications volume 9, Article number: 4742 (2018)*”.